# Ventral tegmental area interneurons revisited: GABA and glutamate projection neurons make local synapses

Lucie Oriol[1], Melody Chao[1], Grace J Kollman[1], Dina S Dowlat[1], Sarthak M Singhal[1], Thomas Steinkellner[2], Thomas S Hnasko[1,3]*

[1]Department of Neurosciences, University of California, San Diego, San Diego, United States; [2]Institute of Pharmacology, Center for Physiology and Pharmacology, Medical University of Vienna, Vienna, Austria; [3]Research Service VA San Diego Healthcare System, San Diego, United States

*For correspondence:
thnasko@health.ucsd.edu

Competing interest: The authors declare that no competing interests exist.

## eLife Assessment

This manuscript provides **convincing** evidence derived from diverse state-of-the-art approaches to suggest that non-dopaminergic projection neurons in the ventral tegmental area (VTA) make local synapses. These **important** findings challenge the prevailing wisdom that VTA interneurons exclusively form local synaptic contacts and instead reveal that VTA neurons expressing interneuron markers also form long-range projections to forebrain targets such as the cortex, ventral pallidum, and nucleus accumbens. Given the importance of VTA interneurons to many models of VTA-linked behavioral functions, these findings have significant implications for our understanding of the neural circuits underlying reward, motivation, and addiction.

**Abstract** The ventral tegmental area (VTA) contains projection neurons that release the neurotransmitters dopamine, GABA, and/or glutamate from distal synapses. VTA also contains GABA neurons that synapse locally on to dopamine neurons, synapses widely credited to a population of so-called VTA interneurons. Interneurons in cortex, striatum, and elsewhere have well-defined morphological features, physiological properties, and molecular markers, but such features have not been clearly described in VTA. Indeed, there is scant evidence that local and distal synapses originate from separate populations of VTA GABA neurons. In this study, we tested whether several markers expressed in non-dopamine VTA neurons are selective markers of interneurons, defined as neurons that synapse locally but not distally. Challenging previous assumptions, we found that VTA neurons genetically defined by expression of parvalbumin, somatostatin, neurotensin, or Mu-opioid receptor project to known VTA targets including nucleus accumbens, ventral pallidum, lateral habenula, and prefrontal cortex. Moreover, we provide evidence that VTA GABA and glutamate projection neurons make functional inhibitory or excitatory synapses locally within VTA. These findings suggest that local collaterals of VTA projection neurons could mediate functions prior attributed to VTA interneurons. This study underscores the need for a refined understanding of VTA connectivity to explain how heterogeneous VTA circuits mediate diverse functions related to reward, motivation, or addiction.

## Introduction

The ventral tegmental area (VTA) is a central component of the brain's reward circuitry, and a common attribute of addictive drugs is their ability to increase dopamine release from VTA projections (*Lüscher,*

2016; Nestler, 2005). The VTA projects to and receives inputs from many brain structures involved in reward-related behavior, including nucleus accumbens (NAc), ventral pallidum (VP), lateral habenula (LHb), and prefrontal cortex (PFC) (Fields et al., 2007; Morales and Margolis, 2017). The VTA is often simplified as a region containing dopamine (DA) projection neurons and inhibitory GABA 'interneurons' that regulate DA neurons (Johnson and North, 1992; Lüscher and Malenka, 2011; Nestler, 2005). However, VTA neurons are highly heterogeneous. The VTA contains distinct populations of DA neurons that can be segregated by gene expression, projection target, and function (Azcorra et al., 2023; Poulin et al., 2020; Roeper, 2013). GABA-releasing VTA neurons make local intra-VTA synapses (Bayer and Pickel, 1991; Omelchenko and Sesack, 2009), but also project widely outside the VTA, including dense projections to LHb, VP, and VP-adjacent areas of basal forebrain (Kaufling et al., 2010; Oades and Halliday, 1987; Taylor et al., 2014). Glutamate neurons are also prevalent in VTA and overlap with other populations such that ~25% of VTA glutamate neurons co-express a DA marker and ~25% express a GABA marker (Conrad et al., 2024; Ma et al., 2023; Phillips et al., 2022). VTA glutamate neurons release glutamate locally within VTA and from distal axons in medial NAc, PFC, VP, LHb, and elsewhere (Dobi et al., 2010; Gorelova et al., 2012; Hnasko et al., 2012; Root et al., 2014; Taylor et al., 2014; Yamaguchi et al., 2011).

It is now understood that DA signals can induce or correlate with distinct behavioral responses depending on their projection targets (Azcorra et al., 2023; Badrinarayan et al., 2012; de Jong et al., 2019; Faget et al., 2024). This is true also for VTA GABA and glutamate neurons. For example, activating VTA GABA neurons either locally within VTA or from distal processes can drive behavioral avoidance, disrupt reward seeking, or modify opioid reinforcement (Corre et al., 2018; Root et al., 2020; Shields et al., 2021; Soden et al., 2020; Tan et al., 2012; van Zessen et al., 2012; Zhou et al., 2022). On the other hand, stimulation of VTA GABA projections to LHb can be rewarding (Lammel et al., 2015; Stamatakis et al., 2013). Likewise, stimulation of VTA glutamate neurons can drive robust positive reinforcement or behavioral avoidance depending on the projection target and behavioral assay (Root et al., 2018; Root et al., 2014; Wang et al., 2015; Yoo et al., 2016). These responses can depend also on the co-release of distinct transmitters. For example, activation of VTA glutamate projections to NAc drives positive reinforcement through the release of glutamate and avoidance via DA co-release (Warlow et al., 2024; Zell et al., 2020). Thus, VTA neurons can mediate approach or avoidance behaviors through their specific connectivity and neurotransmitter content, and understanding the circuit mechanisms regulating activity in diverse VTA cell types is crucial to understanding the mechanisms by which mesolimbic circuits control motivated behaviors.

Local intra-VTA GABA modulation of VTA output, particularly DA output, may underlie key aspects of behavioral reinforcement. For example, inhibitory inputs to VTA from lateral hypothalamus, bed nucleus of stria terminalis, or VP can drive positive reinforcement and approach behaviors through inhibition of VTA GABA neurons and disinhibition of VTA DA neurons (Faget et al., 2024; Nieh et al., 2015; Soden et al., 2020; Soden et al., 2023). VTA GABA circuits also appear to be critical for the generation of DA reward prediction error signals (Eshel et al., 2015; Keiflin and Janak, 2015). Moreover, drugs of abuse can induce rapid or plastic changes in DA signaling through mechanisms that depend on intra-VTA GABA transmission (Corre et al., 2018; Gomez et al., 2019; Lüscher and Malenka, 2011; Ostroumov and Dani, 2018; Ting-A-Kee and van der Kooy, 2012). Indeed, the observation that Mu-opioid receptor (MOR) agonists directly inhibit non-DA VTA neurons and produce disinhibitory effects on VTA DA neurons (Johnson and North, 1992) helped establish the notion of VTA interneurons into current models of VTA architecture.

Yet there is scant evidence for the existence of VTA GABA interneurons, defined as neurons that make synapses locally within VTA but that do not make distal connections. Interneurons as so defined in cortex, striatum, and other brain regions have characteristic morphological features, physiological properties, and molecular markers (Markram et al., 2004; Pelkey et al., 2017; Tepper et al., 2010). However, no molecular or physiological feature has been described that can clearly distinguish VTA GABA interneurons from GABA projection neurons. Identifying a marker that selectively labels VTA interneurons would enable investigations into distinct roles for VTA interneurons and projection neurons (Bouarab et al., 2019; Paul et al., 2019).

In this study, we first sought to test whether several genes that are expressed in a subset of VTA neurons may be selective for interneurons in VTA. We chose markers that are expressed in non-DA neurons, selectively label interneurons in other brain areas, and/or have been widely presumed to label

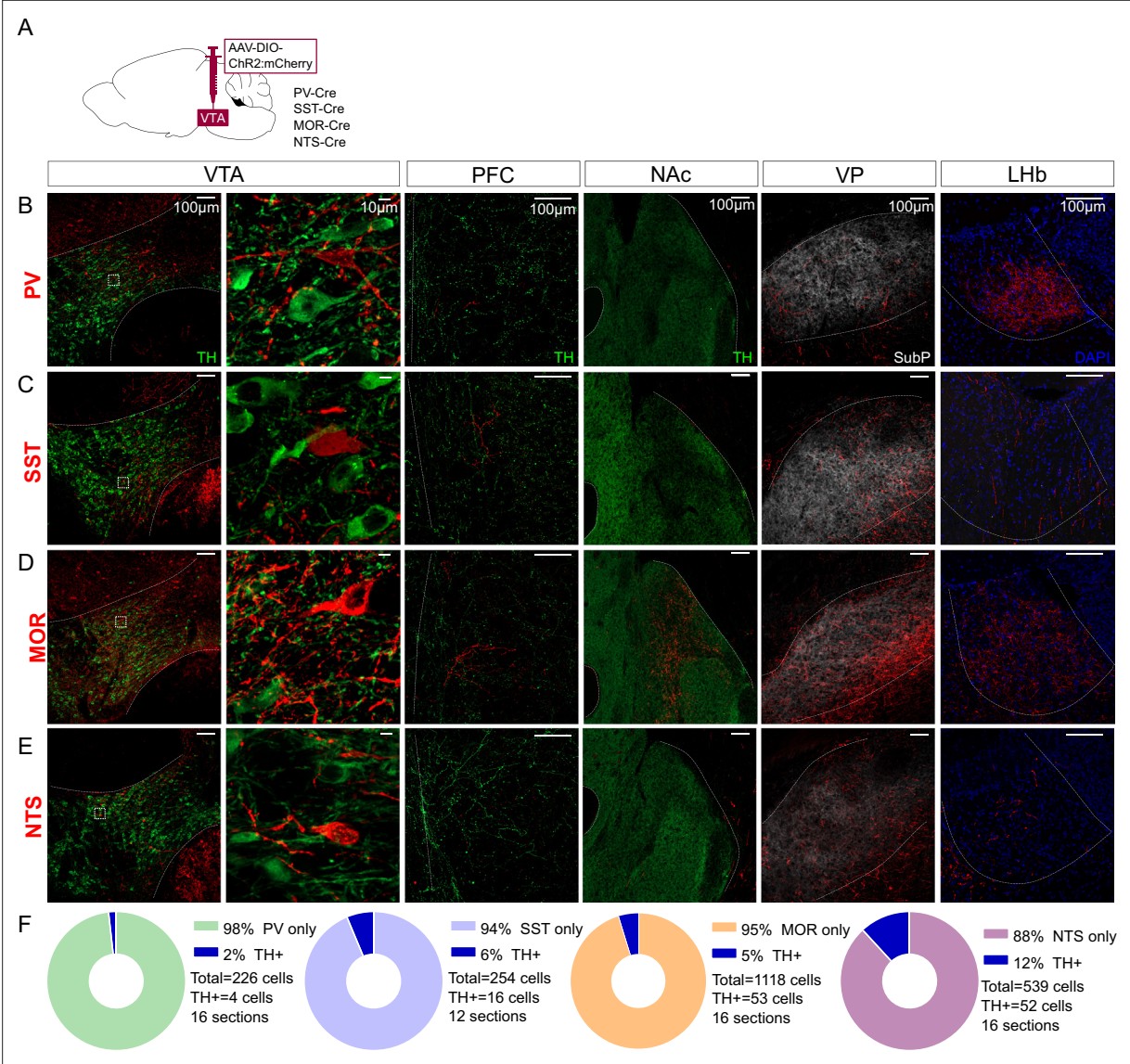

**Figure 1.** Distal projections of putative ventral tegmental area (VTA) interneuron markers. (**A**) Cre-dependent expression of ChR2:mCherry in VTA cell bodies but also distal axonal process in (**B**) PV-Cre, (**C**) SST-Cre, (**D**) MOR-Cre, and (**E**) NTS-Cre mice. First column is an overview of the expression in VTA (bregma –3.3), followed by a high magnification inset of the boxed region in the second column. The third column shows expression patterns in prefrontal cortex (PFC) (bregma +1.7), the fourth in nucleus accumbens (NAc) (bregma +1.3), the fifth in ventral pallidum (VP) (bregma +0.5), and the sixth in lateral habenula (LHb) (bregma –1.8). Scale bars are 100 µm, except 10 µm in the second column. ChR2:mCherry is shown in red; with TH in green, Substance P in white, or DAPI in blue. (**F**) Donut charts show the fraction of mCherry+ VTA cells counted that label for TH.

VTA interneurons. We found that these markers labeled neurons that were primarily non-DA neurons, but that made projections to distinct VTA projection targets, and thus did not selectively label VTA interneurons. We thus sought to test the hypothesis that VTA GABA (or glutamate) projection neurons make intra-VTA collaterals. Indeed, we provide both anatomical and physiological evidence that VTA GABA neurons projecting to NAc, VP, or PFC make local synapses within VTA. This work challenges the presumption of GABA interneurons in VTA by providing direct evidence for an alternative model by which GABA projection neurons can regulate the activity of neighboring VTA cells.

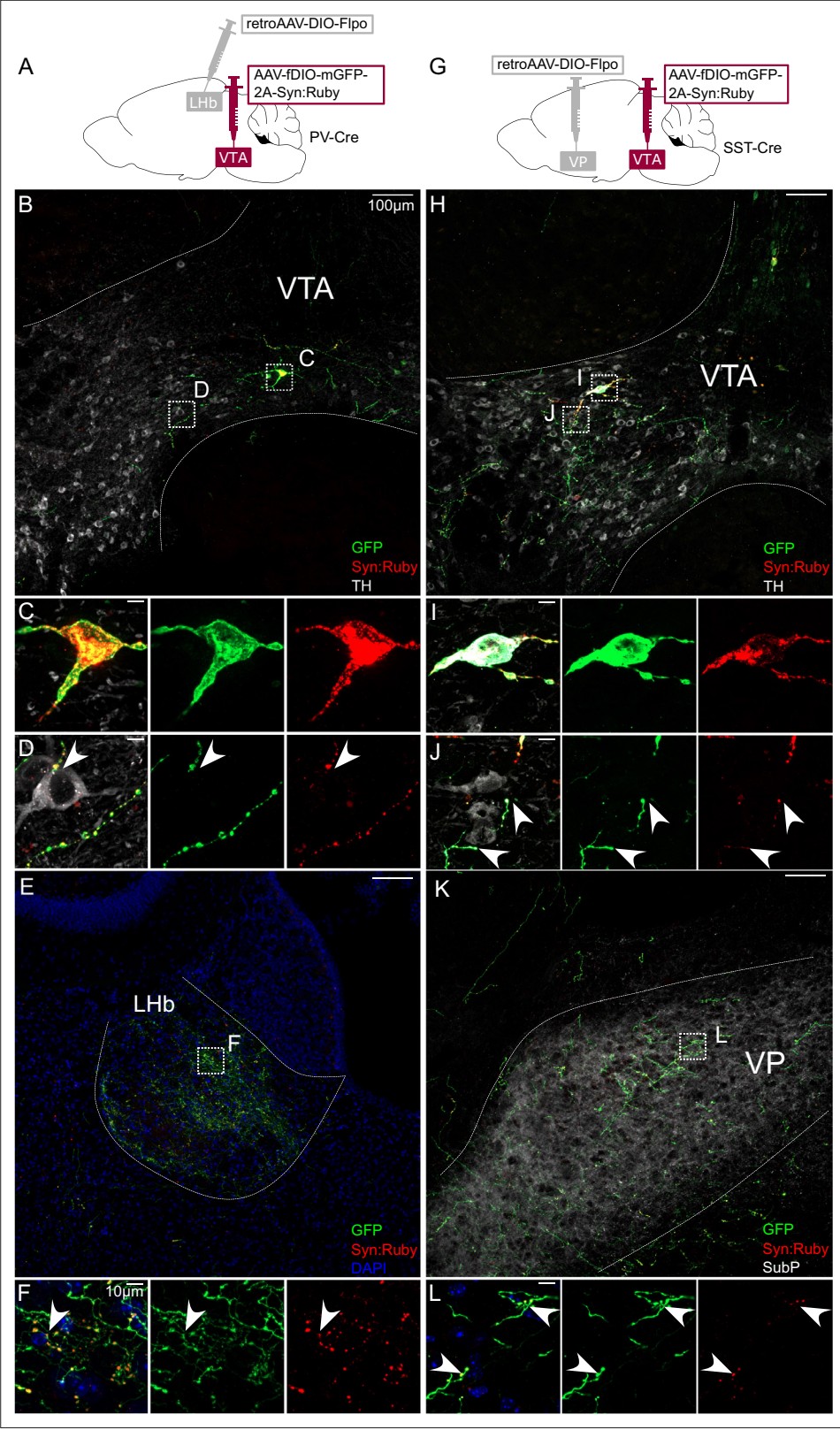

**Figure 2.** Intersectional approach to label projections of PV- and SST-expressing ventral tegmental area (VTA) neurons. (**A**) Dual adeno-associated virus (AAV) approach for Cre-dependent expression of Flp injected in lateral habenula (LHb) plus Flp-dependent expression of GFP and Syn:Ruby in VTA of PV-Cre mice. (**B**) LHb-projecting PV-Cre neurons in VTA with (**C, D**) high magnification insets showing putative release sites proximal to TH+ DA

*Figure 2 continued on next page*

*Figure 2 continued*

neurons. (**E**) VTA axons in LHb with (**F**) high magnification insets. (**G**) Dual AAV approach for Cre-dependent expression of Flp injected in ventral pallidum (VP) plus Flp-dependent expression of GFP and Syn:Ruby in VTA of SST-Cre mice. (**H**) VP-projecting SST-Cre neurons in VTA with (**I,J**) high magnification insets showing putative release sites proximal to TH +DA neurons. (**K**) VTA axons in VP with (**L**) high magnification insets. Scale bars: 100 or 10 μm for high magnification insets.

## Results

### PV, SST, MOR, and NTS are not selective interneuron markers in VTA

We selected four genes with well-validated Cre lines to test as putative genetic markers that might selectively label VTA interneurons: PV-Cre with Cre targeted to the parvalbumin gene, SST-Cre with Cre targeted to the somatostatin gene, NTS-Cre with Cre targeted to the neurotensin gene, or MOR-Cre with Cre targeted to the *Oprm1* gene encoding the MOR. We injected adeno-associated virus (AAV) into the VTA for Cre-dependent expression of Channelrhodopsin-2 (ChR2) fused to mCherry that labels distal axons (***Figure 1A***). To test whether the labeled VTA neurons project distally we assessed expression in known VTA projection sites including NAc, VP, PFC, and LHb.

Parvalbumin (PV) is a marker of interneurons in cortex and striatum (***Kawaguchi, 1993***; ***Kawaguchi and Kondo, 2002***; ***Tepper et al., 2018***), but is also expressed in VTA GABA neurons (***Olson and Nestler, 2007***). Injections into VTA of PV-Cre mice labeled neurons located in medial VTA. We also detected a dense concentration of axonal fibers in LHb, with scant labeling in other known VTA projection targets (***Figure 1B***). These data suggest that PV labels VTA projection neurons and PV is not a selective marker of VTA interneurons.

Like PV, somatostatin (SST) is an interneuron marker in cortex (***Kawaguchi and Kondo, 2002***). SST is expressed in VTA GABA neurons that can inhibit neighboring VTA DA neurons (***Nagaeva et al., 2020***). Injections into SST-Cre mice labeled cell bodies in VTA (***Figure 1C***). We again identified axons in distal targets, here with notably dense labeling in VP.

MOR is expressed in VTA GABA neurons, inhibiting GABA release from synapses on to VTA DA neurons, thereby increasing DA neuron firing, and is often described as a marker of VTA interneurons (***Gysling and Wang, 1983***; ***Johnson and North, 1992***; ***Lüscher and Malenka, 2011***; ***Nestler, 2005***; ***Phillips et al., 2022***). Injections into MOR-Cre mice led to labeled neurons throughout VTA, but also labeled axons in PFC, NAc, LHb, and especially VP (***Figure 1D***).

Neurotensin (NTS) is expressed in a subpopulation of VTA GABA neurons (***Phillips et al., 2022***) and NTS can stimulate mesolimbic DA cells through activation of NTS receptor 1 (***Cáceda et al., 2006***; ***Kalivas et al., 1983***). Injections into NTS-Cre mice labeled neurons in VTA, as well as axons in VP, with weaker labeling in other VTA projection sites (***Figure 1E***).

We also stained VTA sections for tyrosine hydroxylase (TH) to estimate the proportion of ChR2:mCherry neurons colocalizing with DA neurons. In all cases only a minority of mCherry-labeled neurons expressed TH, ranging from 2% for PV to 12% for NTS (***Figure 1F***). In total, our data suggest that these four markers label primarily non-DA neurons in VTA, but that none are selective for interneurons, and instead are inclusive of VTA projection neurons.

### Anatomical evidence that VTA projection neurons make local synapses

Each of the markers tested are also expressed in neurons proximal to VTA and our injections led to variable spread to neighboring regions, including interpeduncular nucleus (IPN) and red nucleus. While these regions are not known to project to PFC, NAc, VP, or LHb, we nonetheless aimed to validate the above findings with a secondary approach involving a combination of retrograde labeling and intersectional genetics to target VTA projection neurons. We injected AAV-fDIO-mGFP-Synaptophysin:mRuby into VTA of each Cre line, plus retroAAV-DIO-Flp into a projection target receiving dense innervation. This approach allowed for Cre- plus Flp-dependent expression of both membrane-localized GFP and the synaptic vesicle marker Syn:Ruby (***Lammel et al., 2015***). The intersectionality of this approach allows for precise targeting of VTA projection neurons, and Syn:Ruby highlights putative release sites, either local to or distal from VTA.

Using this approach to label PV-Cre projectors to LHb (***Figure 2A***), or SST-Cre projectors to VP (***Figure 2G***), revealed GFP-positive soma well-restricted within VTA borders delineated by TH

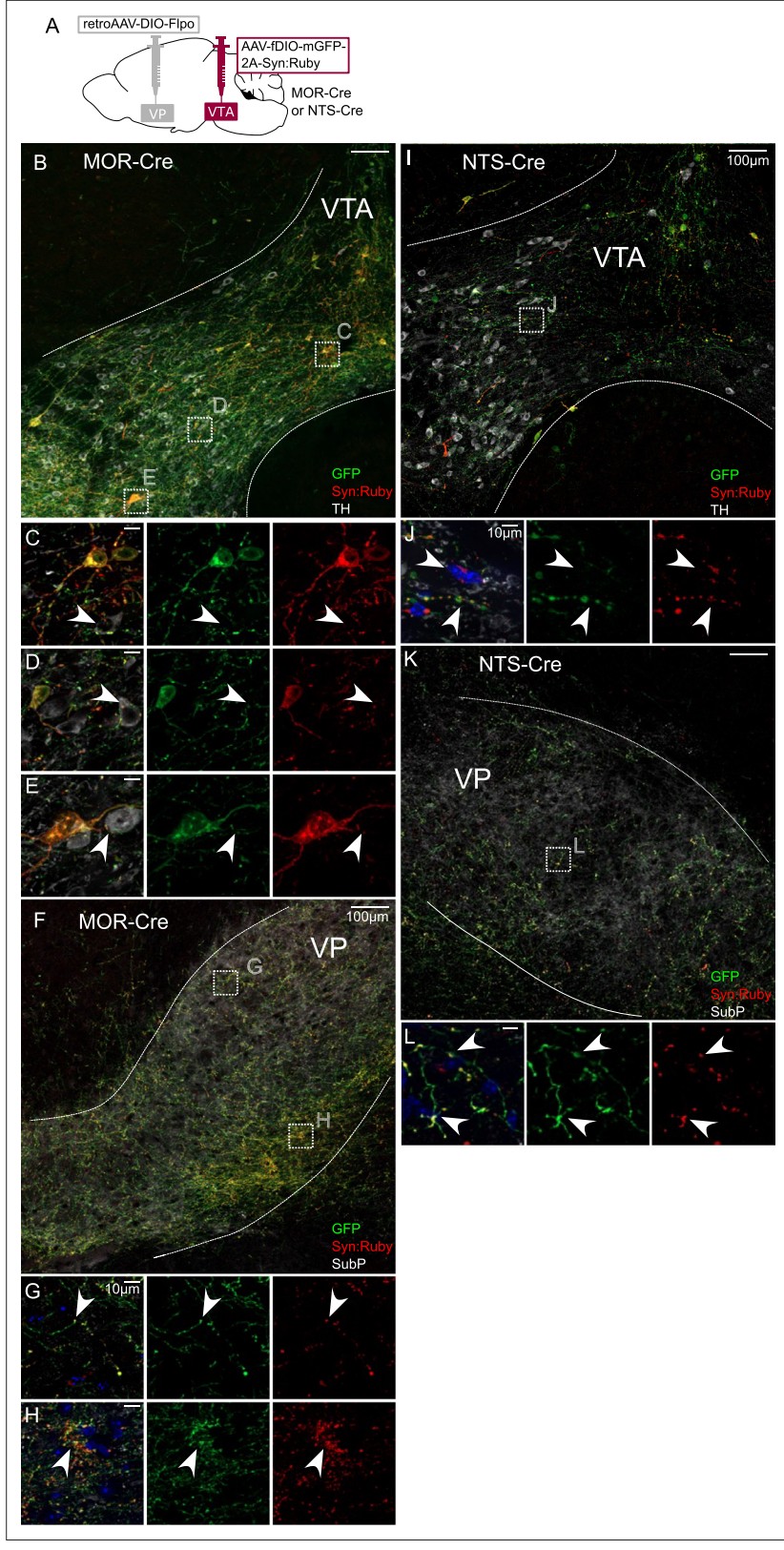

**Figure 3.** Intersectional approach to label projections of Mu-opioid receptor (MOR)- and neurotensin (NTS)-expressing ventral tegmental area (VTA) neurons. (**A**) Dual adeno-associated virus (AAV) approach for Cre-dependent expression of Flp injected in ventral pallidum (VP) plus Flp-dependent expression of GFP and Syn:Ruby in VTA of MOR-Cre and NTS-Cre mice. (**B**) VP-projecting MOR-Cre neurons in VTA with (**C–E**) high magnification

*Figure 3 continued on next page*

*Figure 3 continued*

insets showing putative release sites proximal to TH+ DA neurons. (**F**) VTA axons in VP with (**G, H**) high magnification insets. (**I**) VP-projecting NTS-Cre neurons in VTA with (**J**) high magnification insets showing putative release sites. (**K**) VTA axons in VP with (**L**) high magnification insets. Scale bars: 100 or 10 µm for high magnification insets.

The online version of this article includes the following figure supplement(s) for figure 3:

**Figure supplement 1.** Projection of MOR-Cre-expressing ventral tegmental area (VTA) neurons to ventral pallidum (VP).

immunolabel (*Figure 2B, C, H, I*). We also observed GFP-positive axons and Syn:Ruby-positive puncta in LHb of PV-Cre, or VP of SST-Cre mice (*Figure 2E, F, K, L*). Using high magnification we observed Syn:Ruby puncta proximal to TH-positive cells in VTA (*Figure 2D, J*), suggesting that these VTA projection neurons collateralize within VTA and synapse on to DA neurons.

The same approach was used to label VTA projectors to VP in MOR-Cre or NTS-Cre mice (*Figure 3A*). VP-projecting GFP-positive cell bodies in MOR-Cre mice were contained within and throughout VTA (*Figure 3B*). We also observed axonal fibers densely filling VP, delineated by Substance P immunolabel (*Figure 3F–H* and *Figure 3—figure supplement 1*). Using high magnification, we observed Syn:Ruby puncta proximal to TH-positive VTA neurons, again suggestive of synapses made within VTA by labeled projection neurons (*Figure 3C–E*). Using the same approach to target VP-projecting neurons in NTS-Cre mice we found similar results, with GFP-positive soma in VTA and Syn:Ruby-positive puncta in both VTA (*Figure 3I, J*) and VP (*Figure 3K, L*), though signals were notably less dense. These results corroborate the findings in *Figure 1* and suggest that multiple markers that had been suggested to label putative interneurons instead label VTA projection neurons that may make local synapses through axon collaterals.

There is evidence indicating that PV, SST, MOR, and NTS neurons in VTA express GABA markers or release GABA (*Nagaeva et al., 2020*; *Olson and Nestler, 2007*; *Phillips et al., 2022*). However, some neurons positive for those markers may express VGLUT2 and release glutamate (*Miranda-Barrientos et al., 2021*). We therefore used VGAT-Cre and VGLUT2-Cre mice to selectively express GFP and Syn:Ruby, here targeting NAc-projecting VTA neurons. In VGAT-Cre mice (*Figure 4A*) we identified GFP-positive cell bodies that were restricted to VTA (*Figure 4B*) and GFP-positive fibers in NAc (*Figure 4E, F*). At higher magnification, we observed a pattern of GFP fibers and Syn:Ruby puncta surrounding TH-positive cell bodies (*Figure 4C, D*), suggesting that NAc-projectors make collaterals on to VTA DA neurons. We observed similar results when using VGLUT2-Cre mice, suggesting that NAc-projecting VTA glutamate neurons can also make local collaterals within VTA (*Figure 4G–L*). As expected VTA glutamate cell bodies were concentrated in medial VTA, where they are most dense (*Conrad et al., 2024*; *Kawano et al., 2006*; *Yamaguchi et al., 2011*).

## Physiological evidence that VTA projection neurons make local synapses

Our anatomical results suggest that multiple types of VTA projection neurons collateralize locally within VTA. Next, to functionally assess whether VTA projection neurons make local synapses in VTA, we used a combination of optogenetics and electrophysiology. We selectively expressed ChR2 in NAc-projecting VTA neurons by injecting retroAAV-Cre into NAc and AAV-DIO-ChR2:mCherry into VTA of wild-type mice (*Figure 5A*). We then made acute brain slices to record from VTA neurons negative for ChR2:mcherry to test if they received synaptic inputs from NAc-projecting VTA neurons (*Figure 5B*). Using wild-type mice allowed us to express opsin in both GABA and glutamate projection neurons, and assess for optogenetic-evoked postsynaptic currents (oPSCs) that were either inhibitory (oIPSC) or excitatory (oEPSC) from the same cell.

As expected, the medial shell of NAc showed dense mCherry-positive fibers (*Figure 5C*), and mCherry-positive cell bodies were restricted to VTA (*Figure 5D*). We patched ChR2:mCherry-negative VTA neurons (*Figure 5—figure supplement 1*), flashed 2 ms blue light pulses, and observed oPSCs in 59% of neurons; 44% displayed short-latency oIPSCs (mean 84 ± 17 pA), 6% had short-latency oEPSCs (mean –28 ± 6 pA), 41% had no response (responses less than 5 pA were considered unconnected), and 11% had oPSCs with long latency to onset (>5 ms) (*Figure 5E, F*). Note that unconnected and long-latency cells are not included in *Figure 5F, I*. The GABA$_A$ receptor

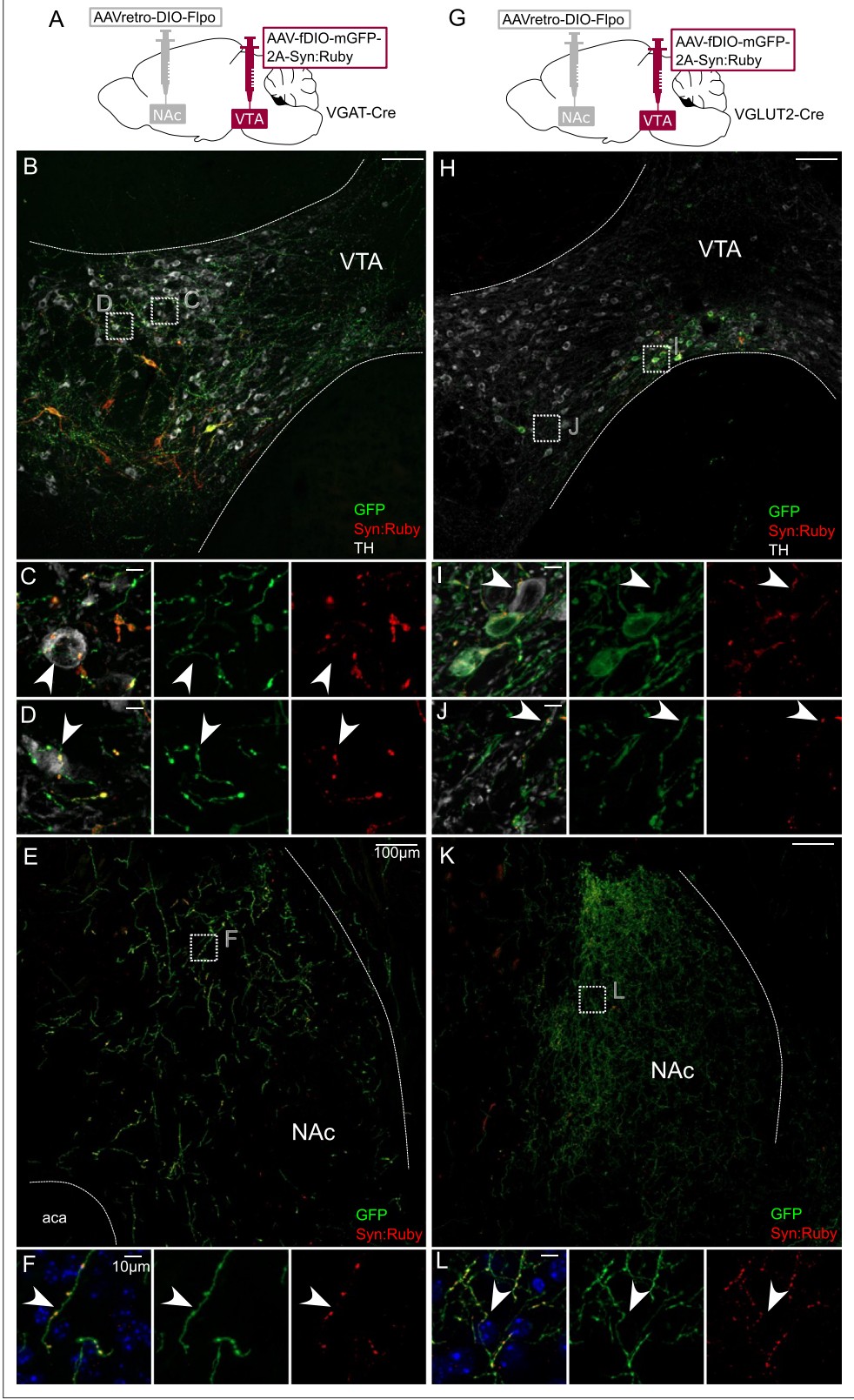

**Figure 4.** Intersectional labeling of ventral tegmental area (VTA) GABA and glutamate projection neurons suggests intra-VTA collaterals. (**A**) Dual adeno-associated virus (AAV) approach for Cre-dependent expression of Flp injected in nucleus accumbens (NAc) plus Flp-dependent expression of GFP and Syn:Ruby in VTA of VGAT-Cre mice. (**B**) NAc-projecting VGAT-Cre neurons in VTA with (**C, D**) high magnification insets showing putative release

*Figure 4 continued on next page*

*Figure 4 continued*
sites proximal to TH+ DA neurons. (**E**) VTA axons in NAc of VGAT-Cre mice, with (**F**) high magnification insets. (**G**) Dual AAV approach for Cre-dependent expression of Flp injected in NAc plus Flp-dependent expression of GFP and Syn:Ruby in VTA of VGLUT2-Cre mice. (**H**) NAc-projecting VGLUT2-Cre neurons in VTA with (**I, J**) high magnification insets showing putative release sites proximal to TH+ DA neurons. (**K**) VTA axons in NAc of VGLUT2-Cre mice with (**L**) high magnification insets. Scale bars: 100 or 10 µm for high magnification insets.

antagonist picrotoxin (PTX) blocked oIPSCs while oEPSCs were blocked by the AMPA receptor antagonist DNQX (*Figure 5G, H*), confirming that these responses are mediated by evoked GABA or glutamate release, respectively.

Most responses displayed onset latencies more than 2 ms and less than 5 ms, consistent with monosynaptic connectivity (3.2 ± 0.1 and 3.8 ± 0.2 ms for oIPSCs and oEPSCs, respectively) (*Figure 5I*). To confirm connections are monosynaptic we performed additional pharmacology. We found that the amplitude of oPSCs was diminished following the application of the voltage-gated sodium channel blocker tetrodotoxin (TTX, voltage-gated sodium channel are necessary for the propagation of action potentials), and that oPSCs recovered with bath application of the inhibitor of voltage-sensitive potassium channels 4-aminopyridine (4AP). When this strategy was applied to oIPSCs (*Figure 5J*), eight out of nine TTX-diminished currents were restored by the application of 4AP (*Figure 5J, K*). Similarly, four of seven oEPSCs were recovered by 4AP (*Figure 5L, M*). We plotted the latency of oPSC onset against the percent oPSC recovery mediated by 4AP and found that 3 of 4 neurons that failed to recover had a latency >5 ms, whereas only 1 of 13 neurons that had a latency <5 ms failed to recover (*Figure 5N*). Therefore, we used 5 ms as a 'short-latency' cutoff to consider an oPSC as monosynaptic. In total we recorded ten cells with oPSC latency >5 ms (identified as long latency in *Figure 5E*). Out of these 10 long-latency oPSCs, 8 were oEPSCs and 2 oIPSCs. This proportion (8:2) of neurons with oEPSCs versus oIPSCs was strikingly greater than that for short-latency responses (6:41), suggesting that in some cells/slices optogenetic stimulation of projection neurons recruited a more extensive intra-VTA excitatory network.

We used a similar approach to assess whether VTA neurons projecting to VP or PFC also made local collaterals in VTA. We used the same combination of viruses but here injected retroAAV-Cre into VP of wild-type mice (*Figure 6A*), again recording from mCherry-negative VTA neurons (*Figure 6B*). As expected, we observed dense mCherry-positive fibers in VP and mCherry-positive cell bodies restricted to VTA (*Figure 6C, D* and *Figure 6—figure supplement 1*). We found that 52% of mCherry-negative neurons were connected (13 of 25), 32% displayed short-latency oIPSCs, 4% had short-latency oEPSCs, and 20% were connected but with long latency (>5 ms) (*Figure 6E–G*). We also patched from postsynaptic neurons in VP and found 89% displayed oPSCs, all with short latency, and as in VTA most currents were inhibitory (*Figure 6H–K*).

Next, we repeated the same approach but for PFC-projecting VTA neurons (*Figure 6L, M*). We observed mCherry-positive fibers in PFC (*Figure 6N*) arising from sparse cell bodies found within the bounds of VTA (*Figure 6O* and *Figure 6—figure supplement 1*). In VTA, we found that 36% of ChR2:mCherry-negative neurons were connected, all of which displayed short-latency oIPSCs (*Figure 6P–R*). Altogether our data indicate that VTA GABAergic projection neurons, and to a lesser extent glutamatergic projection neurons, make functional synapses within VTA.

The use of WT mice in these experiments allowed us to assay for the presence of inhibitory currents mediated by GABA-releasing neurons and for excitatory currents mediated by glutamate-releasing neurons, from the same postsynaptic cells. However, we also performed similar experiments using VGAT-Cre and MOR-Cre mice. In these experiments we used an intersectional approach similar to *Figures 2–4*. We targeted NAc-projecting VGAT-Cre neurons by injecting retroAAV for Cre-dependent expression of Flp into NAc, plus AAV for Flp-dependent expression of ChR2:YFP into VTA (*Figure 7A–D*). We found that the majority of YFP-negative VTA neurons that we recorded displayed short-latency oIPSCs (*Figure 7E–G*) (note that we did not assay for oEPSCs in this experiment).

Finally, we did a similar experiment using MOR-Cre mice to target VP-projecting VTA neurons (*Figure 7H–K*). Here we found that half of the recorded YFP-negative cells were connected, showing primarily short-latency oIPSCs along with fewer short- and long-latency oEPSCs (*Figure 7L–N*).

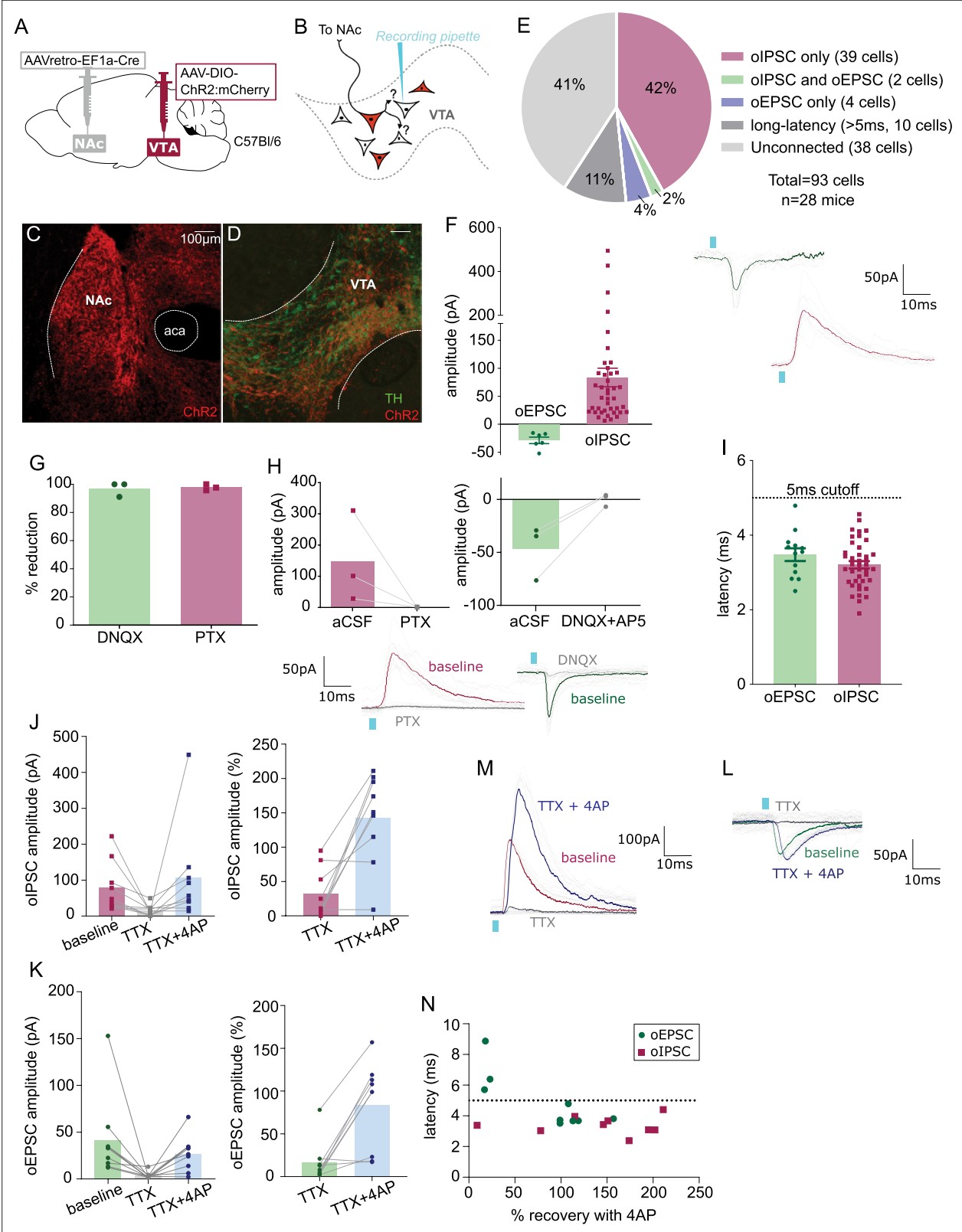

**Figure 5.** Nucleus accumbens (NAc)-projecting ventral tegmental area (VTA) GABA and glutamate neurons make intra-VTA synapses. (**A**) Dual adeno-associated virus (AAV) approach to express ChR2:mCherry in NAc-projecting VTA neurons in wild-type mice. (**B**) Patch-clamp recordings from ChR2:mCherry-negative neurons of VTA to test for collateralizing synapses made by NAc-projectors. (**C**) Coronal images showing ChR2:mCherry expression in NAc and (**D**) VTA; scale bars: 100 μm. (**E**) ChR2:mCherry-negative VTA neuron responses to optogenetic stimulation of NAc-projectors.

*Figure 5 continued on next page*

*Figure 5 continued*

(**F**) Peak amplitude of connected cells that displayed an oEPSC and/or oIPSC (excluding long-latency), with example traces. (**G**) Percent reduction in oEPSC or oIPSC by DNQX or picrotoxin (PTX), respectively. (**H**) Peak amplitude of oIPSCs before and after bath application of PTX, or of oEPSCs before and after bath application of DNQX, with example traces. (**I**) Latency to optogenetic-evoked postsynaptic current (oPSC) onset (excluding long latency). (**J**) Peak oIPSC amplitude before and after bath application of tetrodotoxin (TTX) and recovery with 4-aminopyridine (4AP) (Friedman's test Chi-square = 10.9, p = 0.0029) and (**K**) example traces. (**L**) Peak oEPSC amplitude before and after bath application of TTX and recovery with 4AP (Friedman's test Chi-square = 11.6, p = 0.0013) and (**M**) example traces. (**N**) Scatter plot showing relationship between initial (pre-treatment) latency to oPSC onset and 4AP recovery. Green dots represent oEPSCs and red squares oIPSCs.

The online version of this article includes the following figure supplement(s) for figure 5:

**Figure supplement 1.** Photocurrent and histological validation of approach used in *Figure 5*.

# Discussion

The VTA plays consequential roles in the orchestration of motivated behaviors and is composed of heterogeneous populations of DA, GABA, and glutamate neurons that send dense projections to diverse forebrain regions (*Fields et al., 2007*; *Morales and Margolis, 2017*). Yet VTA DA, GABA, and glutamate neurons also release their neurotransmitters locally within VTA. DA is released from somatodendritic compartments, activating DA autoreceptors (*Ford, 2014*). Multiple lines of evidence indicate that GABA and glutamate neurons resident to VTA synapse locally on to VTA DA and non-DA neurons (*Beier, 2022*; *Dobi et al., 2010*; *Omelchenko and Sesack, 2009*; *Soden et al., 2020*; *Tan et al., 2012*). VTA GABA neurons that make local synapses within VTA have frequently been described as interneurons (*Bonci and Williams, 1996*; *Johnson and North, 1992*; *Lüscher and Malenka, 2011*; *O'Brien and White, 1987*). But there is scant evidence that VTA GABA interneurons and projection neurons represent distinct cell types. In this study, we used retroAAV to target VTA neurons that project to NAc, VP, PFC, or LHb for recombinase-dependent expression of synaptic tags to image putative release sites, or of opsin for optogenetic stimulation of projection neurons while recording synaptic events in neighboring VTA neurons. Both approaches point to the same conclusion, that at least a subset of GABA and glutamate projection neurons collateralize locally and make intra-VTA synapses.

In cortex, hippocampus and other areas dominated by glutamate projection neurons the term interneuron is often used to describe inhibitory GABA neurons that synapse on to neurons within the same structure as their soma reside. However, the limbic basal ganglia circuits in which VTA neurons are embedded include many GABAergic projection neurons, at least subsets of which are understood to make both distal and local synapses. For example, DA D2 receptor-expressing medium spiny neurons are GABA projection neurons that also make extensive local collaterals that laterally inhibit and regulate other striatal neurons (*Dobbs et al., 2016*; *Tunstall et al., 2002*). Thus, a meaningful definition of the term in the context of mesolimbic circuitry, and the definition of interneuron we use in this study, is a neuron that synapses locally but not distally. Indeed, this definition captures many well characterized populations of neurons throughout the cortex, hippocampus, striatum, or cerebellum (*Lim et al., 2018*; *Maccaferri and Lacaille, 2003*; *Pelkey et al., 2017*; *Ascoli et al., 2008*). For example, cortical or striatal interneurons that express PV, SST, or cholinergic markers can be readily distinguished at the molecular level, but also by physiological properties that distinguish them from projection neurons or other cell types (*Huang and Paul, 2019*; *Markram et al., 2004*; *Tepper et al., 2018*).

The VTA has been known to contain non-DA GABA neurons since at least the early 1980s (*Gysling and Wang, 1983*; *Oertel and Mugnaini, 1984*; *Waszczak and Walters, 1980*; *Yim and Mogenson, 1980*). Moreover, VTA GABA neurons were demonstrated to make inhibitory synapses on to VTA DA neurons (*Bayer and Pickel, 1991*; *Johnson and North, 1992*). VTA DA neurons may be distinguished from non-DA neurons (at least in lateral VTA) based on firing rate and other physiological and pharmacological features (*Bunney et al., 1973*; *German et al., 1980*; *Waszczak and Walters, 1980*). These observations serve as the primary basis for the notion of a VTA interneuron. However, those observations could instead be explained by VTA GABA projection neurons that collateralize locally. Indeed, one notable study used in vivo electrophysiology to identify a population of non-DA projection neurons and showed that they were reliably activated by antidromic stimulation of the internal capsule (*Steffensen et al., 1998*). This suggests that the population of non-DA VTA neurons they were able to identify through in vivo recordings were projection neurons. Likewise, substantia nigra

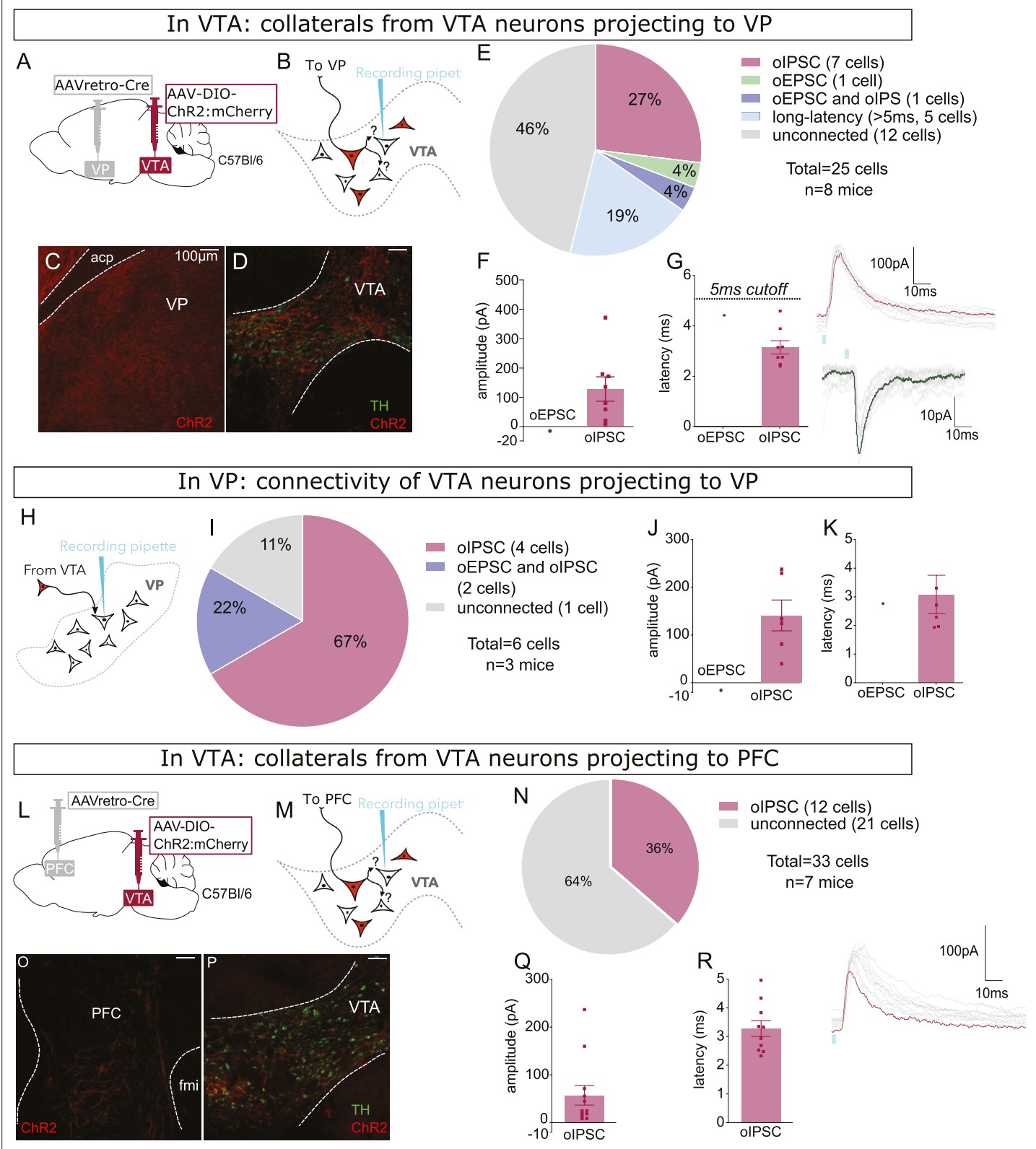

**Figure 6.** Ventral pallidum (VP)- and prefrontal cortex (PFC)-projecting ventral tegmental area (VTA) GABA and glutamate neurons make intra-VTA synapses. (**A**) Dual adeno-associated virus (AAV) approach to express ChR2:mCherry in VP-projecting VTA neurons in wild-type mice. (**B**) Patch-clamp recordings from ChR2:mCherry-negative neurons of VTA to test for collateralizing synapses made by VP-projectors. (**C**) Coronal images showing ChR2:mCherry expression in VP and (**D**) VTA; scale bars: 100 µm. (**E**) ChR2:mCherry-negative VTA neuron responses to optogenetic stimulation of VP-

*Figure 6 continued on next page*

*Figure 6 continued*

projectors. (**F**) Peak amplitude and (**G**) onset latency of connected cells that displayed an oEPSC and/or oIPSC (excluding long latency), with example traces. (**H**) Recordings of optogenetic-evoked postsynaptic currents (oPSCs) from neurons in VP and (**I**) VP responses to optogenetic stimulation of VP-projecting VTA neurons from approach described in panel A. (**J**) Peak amplitude and (**K**) onset latency of connected VP neurons that displayed an oEPSC and/or oIPSC, with example traces. (**L**) Dual AAV approach to express ChR2:mCherry in PFC-projecting VTA neurons in wild-type mice. (**M**) Patch-clamp recordings from ChR2:mCherry-negative neurons of VTA to test for collateralizing synapses made by PFC-projectors. (**N**) Coronal images showing ChR2:mCherry expression in PFC and (**O**) VTA; scale bars: 100 µm. (**P**) ChR2:mCherry-negative VTA neuron responses to optogenetic stimulation of PFC-projectors. (**Q**) Peak amplitude and (**R**) onset latency of connected cells that displayed an oEPSC and/or oIPSC, with example trace.

The online version of this article includes the following figure supplement(s) for figure 6:

**Figure supplement 1.** Histological validation of approach used in *Figure 6*.

(SN) compacta DA neurons were inhibited by antidromically identified GABA projection neurons in SN reticulata (*Tepper et al., 1995*), suggesting a parallel between our findings in VTA and those in SN.

If GABA interneurons represent one or more bona fide VTA cell types, then it is reasonable to suppose that they would be distinguishable by a molecular marker (or a constellation of markers). Markers that confer a GABAergic identity label VTA GABA projection neurons, and thus unlikely to distinguish putative interneurons from projection neurons. However, several other markers have been shown to co-localize with a subset of VTA GABA but not DA neurons and thus represent potential interneuron markers (*Bouarab et al., 2019*; *Nagaeva et al., 2020*; *Olson and Nestler, 2007*; *Phillips et al., 2022*). We selected four of these markers that had well-validated Cre lines: PV, SST, MOR, and NTS. Using two different tracing strategies we confirmed that these markers are expressed in a subset of VTA neurons that are primarily non-dopaminergic. We found that PV$^+$ VTA neurons project densely to LHb (and weakly to other projection targets), while SST$^+$, NTS$^+$, and especially MOR$^+$ VTA neurons project densely to VP (and other projection targets). Thus, while it is possible that there exists VTA interneurons that express one or more of these markers, none of these markers can be used on its own to discriminate between VTA projection neurons and VTA interneurons.

While the retroAAV approach resulted in strong labeling of projection neurons, including GABA- and glutamate-releasing neurons, it is likely that the intrinsic tropism of retroAAV influenced the population of projection cells that we labeled. Indeed, prior work showed that midbrain DA neurons are not efficiently targeted by the retroAAV vector we used (*Tervo et al., 2016*). Thus, populations of neurons that release DA and co-release GABA or glutamate may not contribute to the signals we measured.

In addition to GABA projection neurons, our experiments revealed that glutamate projection neurons in VTA also make local synapses. Therefore, the local excitatory synaptic events observed in prior studies (*Dobi et al., 2010*; *McGovern et al., 2023*; *Yoo et al., 2016*) may be driven by collaterals made by VTA glutamate projection neurons rather than a population of glutamate interneurons. Interestingly, we found that optogenetic activation of unspecified VTA projection neurons induced intra-VTA oEPSCs more rarely than intra-VTA oIPSCs. However, we also observed long-latency oPSCs, that were likely the result of activating VTA glutamate projection neurons that make intra-VTA excitatory collaterals and drive feed-forward recruitment of other VTA cells that also make local synapses.

The VTA integrates a large number of inhibitory inputs from a multitude of brain regions. However, recent studies indicate that neurons local to VTA preferentially inhibit DA neurons compared to GABAergic afferents from distal sources (*Beier, 2022*; *Soden et al., 2020*). Yet these studies cannot determine whether the local neurons synapsing on to VTA DA neurons are interneurons versus collaterals made by projection neurons. Moreover, VTA has been shown to receive an important GABAergic input from the rostral medial tegmental nucleus (RMTg) (*Jhou, 2005*; *Perrotti et al., 2005*). While VTA and RMTg are considered separate structures, the boundary between caudal VTA and rostral RMTg is ambiguous, and this area is dominated by GABA neurons (*Smith et al., 2019*). Interestingly, RMTg inhibitory synapses on to VTA DA neurons are strongly inhibited by MOR activation, and thus some of the functions classically attributed to VTA interneurons may be mediated by these short-range projection neurons (*Jhou, 2021*; *Jhou et al., 2009*; *Kaufling and Aston-Jones, 2015*; *Matsui et al., 2014*; *St Laurent et al., 2020*).

Within VTA, we found that MOR and several other potential interneuron markers were instead expressed in projection neurons. Other interneuron marker candidates have been suggested but, to our knowledge, no other VTA marker has been shown to be expressed selectively within a VTA interneuron population (*Bouarab et al., 2019*; *Paul et al., 2019*). One promising candidate is neuronal nitric

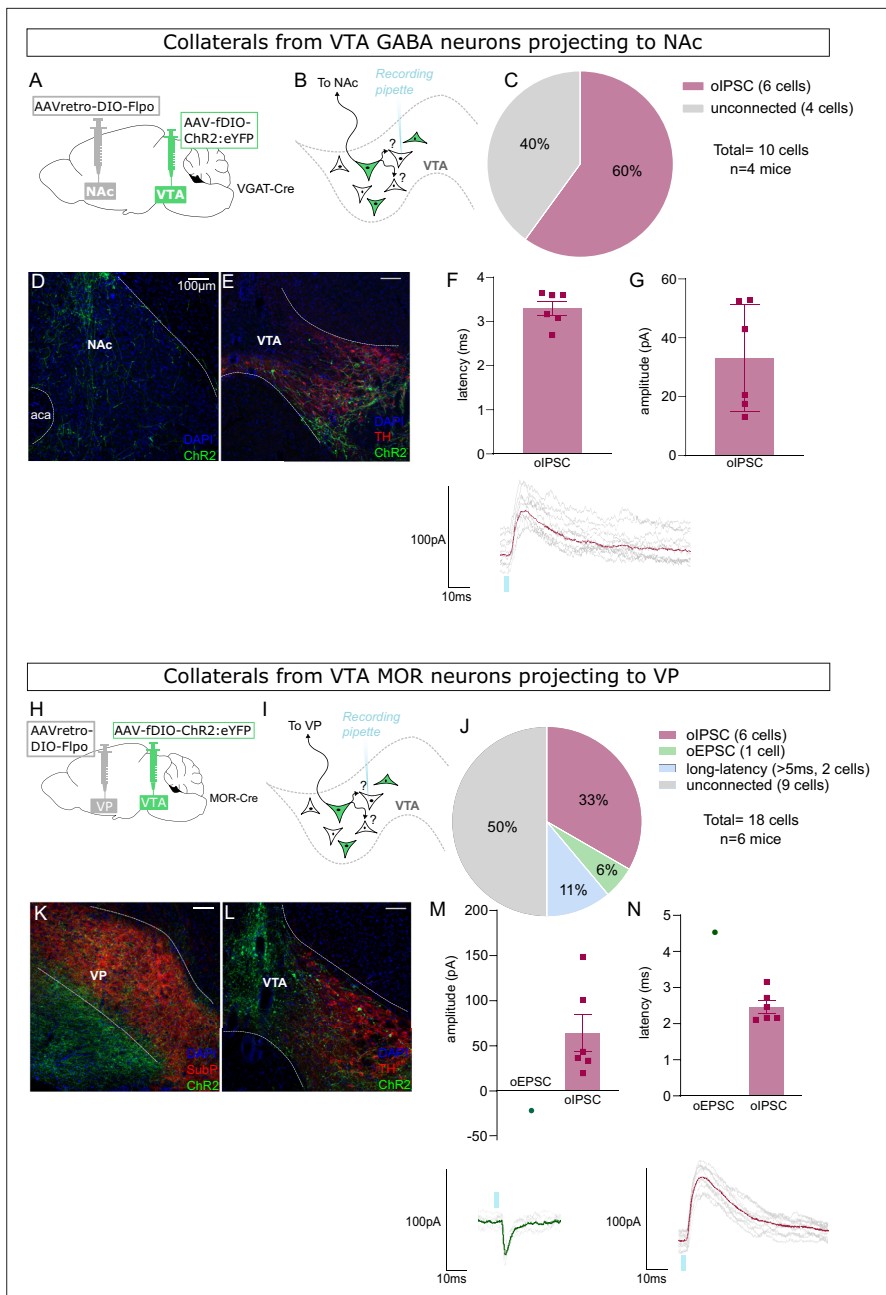

**Figure 7.** Nucleus accumbens (NAc)-projecting ventral tegmental area (VTA) GABA neurons, and ventral pallidum (VP)-projecting VTA Mu-opioid receptor (MOR) neurons, make intra-VTA synapses. (**A**) Dual adeno-associated virus (AAV) approach to express ChR2:eYFP in NAc-projecting VTA neurons in VGAT-Cre mice. (**B**) Patch-clamp recordings from ChR2:eYFP-negative neurons of VTA to test for collateralizing synapses made by NAc-projectors. (**C**) Coronal images showing ChR2:eYFP expression in NAc and (**D**) VTA; scale bars: 100 µm. (**E**) ChR2:eYFP-negative VTA neuron responses to optogenetic stimulation of NAc-projectors. (**F**) Peak amplitude of connected cells that displayed an oIPSC, with example trace. (**G**) Latency to oIPSC onset. (**H**) Dual AAV approach to express ChR2:eYFP in VP-projecting VTA neurons in MOR-Cre mice. (**I**) Patch-clamp recordings from ChR2:eYFP-negative neurons of VTA to test for collateralizing synapses made by VP-projectors. (**J**) Coronal images showing ChR2:eYFP expression in VP and (**K**) VTA; scale bars: 100 µm. (**L**) ChR2:eYFP-negative VTA neuron responses to optogenetic stimulation of VP-projectors. (**M**) Peak amplitude of connected cells that displayed an oEPSC and/or oIPSC (excluding long latency), with example traces. (**N**) Latency to optogenetic-evoked postsynaptic current (oPSC) onset (excluding long latency).

oxide synthase (nNOS) which labels a subset of VTA GABA neurons in the parabrachial pigmented area of VTA that may not project distally, but also labels DA and glutamate neurons in adjacent areas of VTA and SN (*Paul et al., 2018*). Future work, for example using intersectional labeling, may resolve whether nNOS selectively labels bona fide GABA interneurons in VTA. Another candidate marker of interest is prepronociceptin (PNOC). A recent report showed that PNOC labels a population of paranigral non-DA neurons that make dense intra-VTA synapses without projecting to NAc (*Parker et al., 2019*). However, VTA PNOC neurons express both GABA and glutamate markers, and it is not clear whether they project to other VTA projection targets, such as VP or LHb.

In sum, we provide multiple lines of evidence that VTA GABA (and glutamate) neurons that project to distal targets also collateralize locally and make intra-VTA synapses. We also demonstrate that several candidate markers, including MOR, are expressed in VTA projection neurons. Future efforts may reveal positive evidence for the existence of VTA interneurons, for example through the identification of a marker, or a combinatorial set of markers, that labels VTA neurons that make local but not distal connections. At present, however, there is little evidence to support the notion of a VTA interneuron. We suggest that some functions prior attributed to VTA interneurons, such as MOR-mediated disinhibition of DA neurons, may instead be mediated by VTA projection neurons that make synaptic collaterals on to DA neurons. In this way, the actions of opioids on VTA neurons would not only disinhibit DA neurons, but simultaneously inhibit GABA (or glutamate) release from distal VTA projections to VP and elsewhere. Indeed, in light of our increasing understanding for the roles of VTA GABA and glutamate projections in processes underlying behavioral reinforcement, their direct effects on distal targets may contribute to opioid-induced behaviors or adaptations relevant to drug addiction distinct from their effects on VTA DA neurons.

## Methods

### Animals

Mice were group-housed (up to 5 mice/cage), bred at the University of California, San Diego (UCSD), kept on a 12-hr light–dark cycle, and had access to food and water ad libitum. Initial breeders were acquired from The Jackson Laboratory (*Table 1*), except for the MOR-Cre (*Bailly et al., 2020*) obtained from the lab of Brigitte Kieffer (University of Strasbourg). All mice were bred with a C57Bl/6 background and used as a mix of heterozygotes and homozygotes. Male and female mice were used in all experiments. All experiments were performed in accordance with protocols approved by the UCSD Institutional Animal Care and Use Committee.

### Stereotaxic surgery

Mice >5 weeks (and up to 6 months old) were deeply anesthetized with Isoflurane (502017, Primal Critical Care) and placed on a stereotaxic frame (Kopf 1900) for microinjection into discrete brain areas (*Table 2*). After ensuring the skull is flat small holes were drilled (1911-C Kopf) and AAVs (*Table 3*) infused with Nanoject (3-000-207, Drummond) using glass injectors (3-000-203-G/X, Drummond) pulled on a horizontal pipette puller (P-1000 Sutter Instrument). After infusion the injector was left for 3–5 min then withdrawn. Analgesia was provided via injections with 5 mg/kg S.C. Carprofen (510510 Vet One). Electrophysiology was performed >3 weeks after surgery, histology >5 weeks.

**Table 1.** Mouse lines.

| Gene | Abbreviation | Mouse line | Jackson Labs # |
|---|---|---|---|
| *Slc32a1* | VGAT-Cre | B6J.129S6(FVB)-Slc32a1[tm2(cre)Lowl]/MwarJ | 028862 |
| *Slc17a6* | VGLUT2-Cre | STOCK Slc17a6[tm2(cre)Lowl]/J | 016963 |
| *Pvalb* | PV-Cre | B6.129P2-Pvalb[tm1(cre)Arbr]/J | 017320 |
| *Nts* | NTS-Cre | B6;129-Nts[tm1(cre)Mgmj]/J | 017525 |
| *Sst* | SST-Cre | B6N.Cg-Sst[tm2.1(cre)Zjh]/J | 018973 |
| *Gt(Rosa)26Sor* | R26-ZsGreen | B6.Cg-Gt(ROSA)26Sor[tm6(CAG-ZsGreen1)Hze]/J | 007906 |

**Table 2.** Stereotaxic coordinates.

| Injection site | ML | AP | DV |
|---|---|---|---|
| VTA | −0.35 | −3.35 | −4.3 |
| NAc | −0.8 | 1.34 | −4.5 |
| PFC | −0.4 | +1.9 | −1.7 |
| VP | −1.45 | +0.55 | −5.35 |
| VTA (MOR-Cre) | −0.6 | −3.4 | −4.4 |

## Histology

Mice were deeply anesthetized with pentobarbital (200 mg.kg-1, i.p., 200-071, Virbac) and transcardially perfused with 30 ml of PBS (BP399, Fisher bioreagents) followed by 50 ml of 4% PFA (18210, Electron Microscopy Sciences) in PBS. Brains were removed, post-fixed in 4% PFA overnight, and dehydrated in 30% sucrose (S0389, Sigma-Aldrich) in PBS for 48 hr then flash-frozen in isopentane. Brains were cut in 30 µm coronal sections on a cryostat (CM3050S, Leica). Sections were selected to encompass the VTA and efferents to PFC, NAc, VP, and LHb. Sections were blocked in 5% normal donkey serum/0.4% Triton X-100 in PBS for 1 hr at room temperature and incubated with primary antibodies (*Table 4*) overnight at 4°C in the blocking buffer. Next day, slides were washed three times in 0.4% Triton X-100 in PBS for 5 min and incubated with secondary antibodies for 2 hr at room temperature shielded from the light. Finally, sections were washed three times in 0.4% Triton X-100 in PBS for 5 min and coverslipped with Fluoromount-G (Southern Biotech) containing 0.5 µg/ml of DAPI (Roche). Images were taken using a Zeiss Axio Observer Epifluorescence microscope. For *Figure 3— figure supplement 1*, the same procedure was used but the brains were cut sagittaly at a 15° angle.

## Colocalization with TH and counting

For each genetic marker, three to four mice and four sections through VTA per mouse were stained with antibodies against TH and DsRed. All sections were imaged at 10× with the same exposure parameters, using a Zeiss AxioObserver equipped with Apotome2 for structured illumination. The same display settings were applied to all images within condition. TH signal was used to define the boundaries of VTA and align to Bregma point. Cells expressing mCherry were identified first, then scored for presence or absence of TH expression. The counts were done independently by two experimenters and a high correlation was observed between the experimenters ($R^2$ = 0.77, p < 0.001, 60 total sections). Each cell that was only identified by one observer was reassessed for inclusion in final dataset.

## Single injection tracing

For evaluation of projection targets following a single AAV injection into VTA, we excluded subjects that had <30% of labeled cell bodies outside the VTA (*Table 5*). We also excluded subjects that had mCherry-labeled cell bodies in supramammillary nucleus. But we did not exclude mice with spread to red nucleus or IPN because these regions are not known to project to NAc, PFC, VP, or LHb (*Liang et al., 2011*; *McLaughlin et al., 2017*).

**Table 3.** AAV vectors.

| AAV | Titer | Packaged by | Volume | Addgene # |
|---|---|---|---|---|
| AAVretro-EF1a-Cre | 3 × 10[13] | Salk GT3 | 150 nl | 55636 |
| AAV5-EF1α-DIO-hChR2(H134R)-mCherry | 2 × 10[13] | Addgene | 150 nl for ephys 100 nl for histology | 20297 |
| AAVDJ-hSyn1-FLExFRT mGFP-2A-Synaptophysin:mRuby | 2 × 10[13] | Addgene | 150 nl | 71761 |
| AAVretro-hSyn1-DIO-Flpo | 2 × 10[12] | Salk GT3 | 150 nl | NA |

**Table 4.** Antibodies.

| Primary antibody | Species | Catalog # | Company | Dilution |
|---|---|---|---|---|
| TH | Sheep | P60101 | Pel-Freez | 1:2000 |
| DsRed | Rabbit | 632496 | Clontech | 1:2000 |
| GFP | Chicken | A10262 | Invitrogen | 1:2000 |
| Substance P | Rat | MAB356 | Millipore | 1:200 |
| Chat | Goat | AB144P | Millipore | 1:400 |

| Donkey secondary antibody | Alexa Fluor conjugate | Catalog # | Company | Concentration |
|---|---|---|---|---|
| anti-Sheep | 488 | 713-545-003 | | |
| anti-Sheep | 594 | 713-585-147 | | |
| anti-Sheep | 647 | 713-605-147 | | |
| anti-Rabbit | 594 | 711-585-152 | | |
| anti-Chicken | 488 | 703-546-155 | | |
| anti-Rat | 647 | 712-605-153 | | |
| anti-Goat | 647 | 705-605-147 | Jackson Immuno Research | 3 µg/ml |

## Electrophysiology

Mice were deeply anesthetized using pentobarbital (200 mg/kg, i.p., Virbac) and transcardially perfused with 30 ml cold $N$-methyl-D-glucamine (NMDG)-artificial Cerebro-Spinal Fluid (aCSF, containing in mM: 92 NMDG, 2.5 KCl, 1.25 $NaH_2PO_4$, 30 $NaHCO_3$, 20 HEPES, 25 D-glucose, 5 sodium ascorbate, 2 thiourea, 3 sodium pyruvate, 10 $MgSO_4$, 0.5 $CaCl_2$, pH 7.3) saturated with carbogen (95% $O_2$–5% $CO_2$). Sections (coronal, 200 µm) were cut through VTA while immersed in cold NMDG-aCSF using a vibratome (VT1200S, Leica). Slices were incubated at 33°C for 25–30 min in a holding chamber containing NMDG-aCSF saturated with carbogen. During the incubation NaCl concentration was slowly increased in 5 min increments by spiking the holding-aCSF with a 2 M NaCl solution diluted with the NMDG-aCSF (*Ting-A-Kee and van der Kooy, 2012*). Slices were incubated at 25°C for 30–45 min in a holding chamber containing holding-aCSF (containing in mM: 115 NaCl, 2.5 KCl, 1.23 NaH2PO$_4$, 26 $NaHCO_3$, 10 D-glucose, 5 sodium ascorbate, 2 thiourea, 3 sodium pyruvate, 2 $MgSO_4$, 2 $CaCl_2$, pH7.3) saturated with carbogen. While recording, slices were superfused with 31°C recording-aCSF (containing in mM: 125 NaCl, 2.5 KCl, 1.20 $NaH_2PO_4$, 26 $NaHCO_3$, 12.5 D-glucose, 2 $MgSO_4$, 2 $CaCl_2$) using an in-line heater (TC-324B, Warner) at 1.5 ml/min. Whole-cell patch-clamp recordings from mCherry-negative VTA neurons were performed under visual guidance with infrared illumination and differential interference contrast using a Zeiss Axiocam MRm, Examiner.A1 equipped with a ×40 objective. 6–7 MΩ patch pipettes were pulled from borosilicate glass (Sutter Instruments) and filled with internal solution (containing in mM: 133.4 cesium-methanesulfonate, 22.7 HEPES, 0.45 EGTA, 3.2 NaCl, 5.7 tetraethylammonium-chloride, 0.48 NA-GTP, 4.5 $Na_2$-ATP, pH to 7.3 with Cesium-OH). Postsynaptic currents were recorded in whole-cell voltage clamp (Multiclamp 700B amplifier, Axon Instruments), filtered at 2 kHz, digitized at 20 kHz (Axon Digidata 1550, Axon Instruments), and collected using pClamp 10 software (Molecular Device). Neurons were first held at –65 mV to record excitatory currents and then at 0 mV to record inhibitory currents. oPSCs were induced by flashing blue light (two 10 Hz 2 ms pulses, every 15 s) through the light

**Table 5.** Cases included/excluded for *Figure 1*.

| | Surgeries (*n*) | Tracing cases (M/F) | Excluded: spread | Excluded: technical failure | TH counting cases (M/F) |
|---|---|---|---|---|---|
| PV | 9 | 2/1 | 4 | 2 | 3/1 |
| SST | 5 | 0/3 | 2 | 0 | 0/3 |
| MOR | 13 | 3/0 | 8 | 2 | 3/1 |
| NTS | 5 | 0/4 | 1 | 3 | 0/4 |

**Table 6.** Drugs and physiology reagents.

| Reagent | Catalog # | Company |
| --- | --- | --- |
| 4AP | 0940 | Tocris |
| $CaCl_2 \cdot 2H_2O$ | BP510 | Fisher Bioreagents |
| Ces met | 2550-61-0 | Sigma-Aldrich |
| D-Glucose | G8270 | Sigma |
| DNQX | D0540 | Sigma |
| EGTA | E3889 | Sigma |
| HEPES | H3375 | Sigma |
| KCl | BP366 | Fisher Bioreagents |
| $MgSO_4 \cdot 7H_2O$ | M80 | Fisher Bioreagents |
| Na-GTP | G8877 | Sigma |
| $Na_2$-ATP | A2383 | Sigma |
| NaCl | BP358 | Fisher Bioreagents |
| $NaH_2PO_4$ | BP329 | Fisher Bioreagents |
| $NaHCO_3$ | BP328 | Fisher Bioreagents |
| NMDG | M2004 | Sigma-Aldrich |
| PTX | P1675 | Sigma |
| Sodium ascorbate | A7631 | Sigma |
| Sodium pyruvate | P2256 | Sigma-Aldrich |
| TEA chloride | 86616 | Fluka |
| Thiourea | T8656 | Sigma-Aldrich |
| TTX | 1069 | Tocris |

path of the microscope using a light-emitting diode (UHP-LED460, Prizmatix, 50 mW) under computer control. We discarded likely ChR2+ cells, displaying photocurrent (*Figure 5—figure supplement 1*), identified as starting within 1 ms of the light pulse, as well as cells where the series resistance varied by more than 20%. After breaking-in we waited 2–3 min before beginning optogenetic stimulation. For each cell we first recorded a baseline period (4–6 min) and for some cells baseline was followed by 4–6 min bath application of drug: 1 µM TTX, 50–100 µM 4AP, 10 µM DNQX, 100 µM PTX (*Table 6*). For each condition we averaged the last 10 sweeps; amplitude represented the peak current, and latency calculated as the duration from light onset to current onset.

## Statistics

Data values are presented as means ± SEM. Effects of drug application were subjected to Friedman's test (nonparametric ANOVA) followed by a Dunn's post hoc test. Statistical significance was set at $p < 0.05$.

## Acknowledgements

This work was supported by funds from the National Institutes of Health (R01DA036612) and Veterans Affairs (I01BX005782). We thank Karl Deisseroth and Liqun Luo for AAV vectors provided through Addgene (see methods), and Sarah Uran for technical assistance.

# Additional information

## Funding

| Funder | Grant reference number | Author |
| --- | --- | --- |
| National Institute on Drug Abuse | R01DA036612 | Thomas S Hnasko |
| Veterans Affairs San Diego Healthcare System | I01BX005782 | Thomas S Hnasko |

The funders had no role in study design, data collection, and interpretation, or the decision to submit the work for publication.

## Author contributions

Lucie Oriol, Conceptualization, Data curation, Formal analysis, Supervision, Validation, Investigation, Visualization, Writing – original draft, Project administration, Writing – review and editing; Melody Chao, Investigation, Visualization, Writing – review and editing; Grace J Kollman, Dina S Dowlat, Investigation, Writing – review and editing; Sarthak M Singhal, Formal analysis, Investigation, Writing – review and editing; Thomas Steinkellner, Resources, Writing – review and editing; Thomas S Hnasko, Conceptualization, Resources, Supervision, Funding acquisition, Writing – original draft, Project administration, Writing – review and editing

## Author ORCIDs

Lucie Oriol ⓘ https://orcid.org/0009-0009-2966-0911
Thomas S Hnasko ⓘ https://orcid.org/0000-0001-6176-8513

## Ethics

All experiments were performed in accordance with protocols approved by the UCSD Institutional Animal Care and Use Committee (protocol S12080).

Reviewer #1 (Public review): https://doi.org/10.7554/eLife.100085.3.sa1
Reviewer #2 (Public review): https://doi.org/10.7554/eLife.100085.3.sa2
Reviewer #3 (Public review): https://doi.org/10.7554/eLife.100085.3.sa3
Author response https://doi.org/10.7554/eLife.100085.3.sa4

# Additional files

## Supplementary files

MDAR checklist

## Data availability

Source data consists of digital image files from histological samples and whole-cell electrophysiology recordings and is available at https://doi.org/10.5281/zenodo.15042008.

The following dataset was generated:

| Author(s) | Year | Dataset title | Dataset URL | Database and Identifier |
| --- | --- | --- | --- | --- |
| Hnasko TS, Oriol L | 2025 | Dataset for: Ventral tegmental area interneurons revisited: GABA and glutamate projection neurons make local synapses | https://doi.org/10.5281/zenodo.15042008 | Zenodo, 10.5281/zenodo.15042008 |

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
