## [Editor Report · eLife Assessment]

This manuscript provides **convincing** evidence derived from diverse state-of-the-art approaches to suggest that non-dopaminergic projection neurons in the ventral tegmental area (VTA) make local synapses. These **important** findings challenge the prevailing wisdom that VTA interneurons exclusively form local synaptic contacts and instead reveal that VTA neurons expressing interneuron markers also form long-range projections to forebrain targets such as the cortex, ventral pallidum, and nucleus accumbens. Given the importance of VTA interneurons to many models of VTA-linked behavioral functions, these findings have significant implications for our understanding of the neural circuits underlying reward, motivation, and addiction.

---

## [Referee Report · Reviewer #1 (Public review)]

The manuscript by Lucie Oriol et al. revisits the understanding of interneurons in the ventral tegmental area (VTA). The study challenges the traditional notion that VTA interneurons exclusively form local synapses within the VTA. Key findings of the study indicate that VTA GABA and glutamate projection neurons also make local synapses within the VTA. This evidence suggests that functions previously attributed to VTA interneurons could be mediated by these projection neurons.

The study tested four genetic markers-Parvalbumin (PV), Somatostatin (SST), Mu-opioid receptor (MOR), and Neurotensin (NTS)-to determine if they selectively label VTA interneurons. The findings indicate that these markers label VTA projection neurons rather than selectively identifying interneurons. Using a combination of anatomical tracing and brain slice physiological recordings, the study demonstrates that VTA projection neurons make functional inhibitory or excitatory synapses locally within the VTA. These data challenge the conventional view that VTA GABA neurons are purely interneurons and suggests that inhibitory projection neurons can serve functions previously attributed to VTA interneurons. Thus, some functions traditionally ascribed to interneurons may be carried out by projection neurons with local synapses. This has significant implications for understanding the neural circuits underlying reward, motivation, and addiction.

---

## [Referee Report · Reviewer #2 (Public review)]

Summary:

In this manuscript, authors use a combination of transgenic animals, intersectional viruses, retrograde tracing, and ex-vivo slice electrophysiology to show that VTA projections neurons synapse locally. First, the authors injected a cre-dependent channelrhodopsin into the VTA of PV, SST, MOR, and NTS-Cre mice. Importantly, PV, SST, MOR, and NTS are molecular markers previously used to describe VTA interneurons. Imaging of known VTA target regions identified that these neurons are not localized to the VTA and instead project to the PFC, NAc, VP, and LHb. Next, the authors used an intersectional viral strategy to label projections neurons with both GFP (membrane localized) and Syn:Ruby (release sites). These experiments identified that VTA projection neurons also make intra-VTA synapses. Finally, the authors use a combination of optogenetics and ex-vivo slice electrophysiology to show that neurons projecting from the VTA to the NAc/VP/PFC also synapse locally. Overall, the conclusions are well supported by the data.

Strengths:

Previous literature has described Pvalb, Sst, Oprm1, and Nts as selective markers of VTA interneurons. Here, the authors make use of cre driver lines to show that neurons defined by these genes are not classically-defined interneurons and project to known VTA target regions. Additionally, the authors convincingly use intersectional viral approaches and slice electrophysiology to show that projection neurons synapse onto neighboring cells within the VTA

---

## [Referee Report · Reviewer #3 (Public review)]

Summary:

This study from Oriol et al. first uses transgenic animals to examine projection targets of specific subtypes of VTA GABA neurons (expressing PV, SST, MOR, or NTS). They follow this with a set of optogenetic experiments showing that VTA projection neurons (regardless of genetic subtype) make local functional connections within the VTA itself. Both of these findings are important advances in the field. Notably, both GABAergic and glutamatergic neurons in the VTA likely exhibit these combined long/short-range projections.

Strengths:

The main strength of this study is the series of optogenetic/electrophysiological experiments that provide detailed circuit connectivity of VTA neurons. The long-range projections to the VP (but not other targets) are also verified to have functional excitatory and inhibitory components. Overall, the experiments are well executed and the results are very relevant in light of the rapidly growing knowledge about the complexity and heterogeneity of VTA circuitry.

Another strength of this study is the well-written and thoughtful discussion regarding the current findings in the context of the long-standing question of whether the VTA does or does not have true interneurons.

Comments on revisions:

The authors have addressed all of my questions admirably, and the final result is considerably improved and remains a valuable contribution to the field.

---

## [Author Response]

The following is the authors’ response to the original reviews.

**Reviewer #1**:Regarding the manuscript's clarity, the sentence on page 5, "We also stained VTA sections for Tyrosine hydroxylase (TH) to estimate the rate of ChR2 colocalization with DA neurons," reads awkwardly. Removing the word "rate" could improve clarity.

We have made the recommended clarifying edit (page 5, lines 30-31).

Additionally, the anatomical data and findings are largely non-quantitative in nature. However, solid microscopy images are presented to support each claim. Additional quantification would strengthen the paper, specifically the quantification of projection density for each population and the proportion of each subpopulation that projects to their regions of interest.

To rigorously quantify the projection density of each subpopulation would require a level of exhaustivity our study was not designed for. This is because during microscopy we focused efforts on imaging regions containing dense signals but did not exhaustively image regions receiving apparently weak or no input. While we considered including a semi-quantitative table of projection density, based on the data available we could not discriminate with confidence between, e.g., regions recipient of minimal input versus no input from VTA populations. Thus, while we stand by our descriptive statements we do not expand on those further.

The authors should consider discussing the possibility that subpopulations of these cells could still be true interneurons especially if cells were looked at the single neuron level of resolution.

We agree that some of the VTA populations we studied could include subpopulations that are bona fide interneurons. The identification of alternate markers or combinations of markers, or use of single-cell imaging approaches may indeed support this possibility in future. This is discussed in the context of currently available evidence on page 5 lines 32-34, page 11 lines 2-4, page 12 lines 2-11, and page 12 lines 15-16.

Overall, the paper is well-written and important for the field and beyond.

Thank you!

**Reviewer #2:**
Weaknesses:While the authors use several Cre driver lines to identify GABAergic projection neurons, they then use wild-type mice to show that projection neurons synapse onto neighboring cells within the VTA. This does not seem to lend evidence to the idea that previously described "interneurons" are projection neurons that collateralize within the VTA.

We think the use of WT mice is a strength because it allows us to measure both GABA and non-GABA synapses made by VTA projections on to the same cells within VTA. However, we have also done this experiment targeting NAc-projecting VTA VGAT-Cre neurons, and VP-projecting VTA MOR-Cre neurons. Consistent with the WT dataset, we find that these defined projection neurons also make intra-VTA synapses. These data are now included as Figure 7.

More broadly. Our review of the literature finds very little evidence to support the notion of a VTA interneuron as we define it: VTA neurons that makes only local connections. But the absence of evidence need not imply evidence of absence, thus we do not claim that all VTA neurons previously presumed to be interneurons must be projection neurons. We do express confidence in our findings that VTA projection neurons (that include GABA-releasing neurons) make local synapses in VTA. We argue that in the absence of compelling positive evidence for the existence of VTA interneurons, such as a selective marker, “we”, “the field”, should not presume their existence.

Other suggestions:(1) While the authors present evidence that some projection neurons also synapse locally, there is no quantification as to the proportion of each neuronal subtype that collateralizes within the VTA. This would be a useful analysis.

We agree this would be useful information. But our experiments were not designed to answer this question. Indeed, we have not conceived of a feasible method to discriminate between collateralizing and non-collateralizing VTA projection neurons at the single-cell level, thus we do not know how we would calculate such proportions.

(2) There is significant interest in the molecular heterogeneity and spatial topography of the VTA. Additional analyses of the spatial topography of labeled projectors would be useful. For example, knowing if Pvalb+ projection neurons are distributed throughout the VTA or located along the midline would be a useful analysis.

Prior studies and public databases (e.g., Allen brain atlas, GENSAT) allow one to visualize the location of VTA neurons positive for Pvalb and the other markers we investigated (Olson & Nestler, 2007). However, these label the entire population of neurons and thereby include those that project to any of the various projection targets. There are also studies that have used retrograde labeling approaches to map the distribution of labeled VTA cells projecting to one or another target (Beier et al., 2015; Lammel et al., 2008; Margolis et al., 2006). For example, finding that LHb-projecting neurons (a major target of Pvalb+ VTA neurons) are enriched in medial VTA (Root et al., 2014). From this evidence we might infer that Pvalb+ VTA neurons that project to LHb are likely to be medially biased. Future studies may more carefully map the intersection of specific projection targets for each VTA subpopulation.

**Reviewer #3 (Recommendations For The Authors):**
Weaknesses:This study has a few modest shortcomings, of which the first is likely addressable with the authors' existing data, while the latter items will likely need to be deferred to future studies:(1) Some key anatomical details are difficult to discern from the images shown. In Figure 1, the low-magnification images of the VTA in the first column, while essential for seeing what overall section is being shown, are not of sufficient resolution to distinguish soma from processes. A supplemental figure with higher-resolution images could be helpful.

We uploaded a higher resolution file for figure 1.

Also, where are the insets shown in the second column obtained from? There is not a corresponding marked region on the low-magnification images. Is this an oversight, or are these insets obtained from other sections that are not shown?

This was an oversight, we added the corresponding marked region to the low-magnification images.

Lastly, there is a supplemental figure showing the NAc injection sites corresponding to Figure 5, but not one showing VP or PFC injection sites in Figure 6. Why not?

We added a figure with histology examples for the VP and the PFC injection sites as done for Figure 5, included as Supplemental Figure 3.

(2) Because multiple ChR2 neurons are activated in the optogenetic experiments, it is not clear how common is it for any specific projection neuron to make local connections. Are the observed synaptic effects driven by just a few neurons making extensive local collateralizations (while other projection neurons do not), or do most VTA projection neurons have local collaterals? I realize this is a complex question, that may not have an easy answer.

This is a great question but, indeed, we don’t know the answer. As mentioned in response to Reviewer #2, we are not convinced there is a currently feasible way to discriminate between collateralizing and non-collateralizing cells at the single cell level.

(3) There is something of a conceptual disconnect between the early and later portions of this paper. Whereas Figures 1-4 examine forebrain projections of genetic subtypes of VTA neurons, the optogenetic studies do not address genetic subtypes at all. I do realize that is outside of the scope of the author's intent, but it does give the impression of somewhat different (but related) studies being stitched together. For example, the MOR-expressing neurons seem to project strongly to the VP, but it is not addressed whether these are also the ones making local projections. Also, after showing that PV neurons project to the LHb, the opto experiments do not examine the LHb projection target at all.

This too was raised by Reviewer #2. While addressing this question for all the populations we investigated feels redundant, we now include optogenetic data showing that NAc-projecting VTA VGAT-Cre and VP-projecting VTA MOR-Cre neurons also make local collaterals (Figure 7). We think this allows us to connect the two approaches to a greater degree. Based on our findings using a dual virus approach to express Syn:Ruby in each population of VTA projection neuron, we think it very likely that we’d continue to find similar results using optogenetics-assisted slice electrophysiology for each population.

Other suggestions:

(1) I appreciated the extensive and high-quality anatomical figures shown in Figures 2-4. However, the layout was sometimes left-to-right, and sometimes right-to-left, which felt distracting. At some point, the text refers to "Fig. 3KJ", i.e. with the letters being in backward alphabetical order, and Figures 3I and 3L do not appear mentioned anywhere in the main text, leading me to wonder if that text was intended to read "Fig. 3I-L".

Thank you for noting this. We have harmonized the layout of Figures 2-4 and adjusted the in-text Figure call-outs.

Also, the inset in Figure 3J appears to show local collaterals of NTS neurons in the VTA, since there is no soma in that inset. This is interesting, and worth reporting, but is not explained in either the main text or Figure legend.

We added a more complete description in the result section (page 6 line 25-30).

(2) Perhaps I missed it, but I could not find any mention of the intensity of the LED light delivered during the optogenetic experiments. While acknowledging that this can be variable, do the authors have at least a rough range?

We have added this information to the methods, page 17 line 8.

**Editor's Note:**
Should you choose to revise your manuscript, please double check that you have fully reported all statistics including exact p-values wherever possible alongside the summary statistics (test statistic and df) and 95% confidence intervals.

We confirm that we have fully reported all statistics including exact p-values wherever possible alongside the summary statistics (test statistic and df) and 95% confidence intervals.

Note to Editor and Readers

While reanalyzing our data for resubmission, we discovered that some of the short-latency optogenetic evoked postsynaptic currents (oPSCs) we detected were erroneously categorized. Specifically, some VTA cells that showed large outward currents (oIPSCs) when held at 0 mV, also had small inward currents when held at -60 mV. These small inward currents were initially categorized as oEPSCs, suggesting these VTA cells received input from populations of VTA projection neurons that released GABA and/or glutamate. However, the kinetics of these small inward currents were slow and aligned with the within-cell kinetics of the oIPSCs, indicating that these were very likely mediated by GABA_A_ receptors. In one case the opposite was apparent, with a small PSC initially miscategorized as an oIPSC. These miscategorized oEPSCs and oIPSC were presumably detected because our holding potentials were not precisely identical to the reversal potentials for GABA_A_ and AMPA receptors, respectively. For this reason, we removed these 14 oEPSCs and 1 oIPSCs from our analyses in the revised version. The revised dataset suggests that VTA glutamate projection neurons may be less likely to collateralize widely within VTA compared to GABA projection neurons. But, importantly, this correction does not affect any of our conclusions.

Citations:

Beier, K. T., Steinberg, E. E., DeLoach, K. E., Xie, S., Miyamichi, K., Schwarz, L., Gao, X. J., Kremer, E. J., Malenka, R. C., & Luo, L. (2015). Circuit Architecture of VTA Dopamine Neurons Revealed by Systematic Input-Output Mapping. *Cell*, *162*(3), 622-634. https://doi.org/10.1016/j.cell.2015.07.015

Lammel, S., Hetzel, A., Hackel, O., Jones, I., Liss, B., & Roeper, J. (2008). Unique properties of mesoprefrontal neurons within a dual mesocorticolimbic dopamine system. *Neuron*, *57*(5), 760-773. https://doi.org/10.1016/j.neuron.2008.01.022

Margolis, E. B., Lock, H., Chefer, V. I., Shippenberg, T. S., Hjelmstad, G. O., & Fields, H. L. (2006). Kappa opioids selectively control dopaminergic neurons projecting to the prefrontal cortex. *Proc Natl Acad Sci U S A*, *103*(8), 2938-2942. https://doi.org/10.1073/pnas.0511159103

Olson, V. G., & Nestler, E. J. (2007). Topographical organization of GABAergic neurons within the ventral tegmental area of the rat. *Synapse*, *61*(2), 87-95. https://doi.org/10.1002/syn.20345

Root, D. H., Mejias-Aponte, C. A., Zhang, S., Wang, H. L., Hoffman, A. F., Lupica, C. R., & Morales, M. (2014). Single rodent mesohabenular axons release glutamate and GABA. *Nat Neurosci*, *17*(11), 1543-1551. https://doi.org/10.1038/nn.3823